# A Multicenter Network Analysis Examining the Psychological Effects of the COVID-19 Pandemic on Patients with Eating Disorders versus Their Healthy Siblings

**DOI:** 10.3390/jcm11237187

**Published:** 2022-12-02

**Authors:** Paolo Meneguzzo, Alberto De Mico, Pietro Gori, Alessio Ajello, Enrico Ceccato, Mauro Domenico Consolati, Antonio Vita, Alessandra Sala, Paolo Santonastaso

**Affiliations:** 1Department of Neuroscience, University of Padova, 35122 Padova, Italy; 2Padova Neuroscience Center, University of Padova, 35122 Padova, Italy; 3Department of General Psychology, University of Padova, 35122 Padova, Italy; 4Department of Clinical and Experimental Sciences, University of Brescia, 25121 Brescia, Italy; 5Department of Mental Health and Addiction Services, ASST Spedali Civili of Brescia, 25123 Brescia, Italy; 6Vicenza Eating Disorders Center, Mental Health Department, Azienda ULSS8 “Berica”, 36100 Vicenza, Italy

**Keywords:** COVID-19, eating disorder, sibling, posttraumatic, network analysis, restraint

## Abstract

(1) Background: The impact of the COVID-19 pandemic on individuals with eating disorders (EDs) has been recorded all over the world; the traumatic effects of COVID-19 have exacerbated specific and general psychopathologies in those with EDs. Comparing patients’ and their healthy siblings’ responses might help one evaluate whether there are significant differences between healthy individuals and those struggling with EDs in regard to posttraumatic psychological symptoms. (2) Methods: A sample of 141 ED patients and 99 healthy siblings were enrolled in this study in two different centers specializing in ED treatment. All participants completed the posttraumatic stress disorder (PTSD) checklist and an eating and general psychopathological self-report questionnaire. Network analysis was then applied to evaluate the differences between the populations. (3) Results: No significant differences emerged between the network structures despite the significant differences between patients and their healthy siblings in regard to posttraumatic symptoms, eating, and general psychopathology. (4) Conclusion: The complex nature of the interaction between environmental and personal factors should be evaluated further in individuals with EDs due to how they respond to traumatic events, which exacerbate patients’ psychopathology.

## 1. Introduction

Eating disorders (EDs) are severe psychiatric disorders that may significantly impair an individual’s physical health and psychosocial functioning. They often originate in disordered attitudes and behaviors regarding weight, body shape, and nutrition [1]. The onset of EDs usually takes place during adolescence and might persist into adulthood [2,3]. Considering EDs from a neurodevelopmental perspective, different life events might play roles in the development of the disorder [4]—from genetics to environmental factors [5]—and for this reason, studies that include siblings may help one evaluate the roles of genetics and environmental elements and their combination on EDs.

The interaction between neurobiological substrates and environmental events is clearer when examining traumatic events (which are recorded more frequently in individuals with an ED), and this interaction might reveal the roles of environmental and genetic factors [6,7,8]. A growing body of literature has studied the role of traumatic events in developing EDs, even when the actual effects are still not well understood [9,10]. However, different studies have documented the presence of different biological scars in subjects with EDs and personal histories of maltreatment, in line with the hypothesis of a traumatic echo-phenotype of EDs [11,12,13,14]. According to this perspective, the COVID-19 pandemic might represent an interesting traumatic event for patients [15,16], that has been shared by siblings, and this might help researchers better understand individuals’ coping strategies.

The COVID-19 pandemic is an event that led to the activation of several public health measures, including lockdowns and social distancing, which caused a significant change in people’s daily lives from a social, work, and economic perspective [17,18]. People with previous psychiatric conditions have been found to be more vulnerable to the psychological effects of the pandemic’s restrictions [19,20], especially people with EDs [21]. People with EDs reported an increase in disordered eating behaviors, such as concerns for food, exercise, dietary restrictions, and more frequent compensatory behaviors [22,23,24]. Furthermore, COVID-19 has also been found to impact general psychopathology in terms of anxiety, depression, obsessive-compulsive, and posttraumatic stress symptoms [25]. However, while ED-specific symptoms returned to their previous levels after the first general lockdown, general psychopathological aspects persisted, or, as in the case of anxiety, they worsened further [26]. Therefore, it has been hypothesized that individuals with EDs who experience difficulty in regulating emotions may experience exacerbated symptoms in the presence of a traumatic event, such as the pandemic, including social isolation and loneliness [26,27,28]. When studying healthy siblings’ responses, the similarities between patients and the relationships between posttraumatic symptoms and other psychopathological constructs have already been reported in relation to the pandemic’s effects—such as interpersonal sensitivity and obsessive-compulsiveness [29]—calling for further investigation to better understand these results.

Therefore, the aim of this study was to explore the relationships between posttraumatic stress, EDs, and general psychopathology using a network analysis approach on a sample of ED patients and their siblings. This type of analysis allows one to evaluate the relationships between psychological features and compare these relationships between different populations. This approach might be more helpful than standard statistical approaches because it shows the connections between the constructs more than it shows the differences in the variances, and a growing number of studies have applied this methodology in the ED research field [30].

## 2. Materials and Methods

### 2.1. Participants

Participants were recruited between January 2021 and September 2021 from the Eating Disorders Center of the Mental Health Department of AULSS 8, Vicenza (Italy), and the Eating Disorders Center of the Department of Mental Health and Addiction Services of the ASST Spedali Civili of Brescia (Italy). Both facilities are structured as outpatient services with a day hospital ward. The patients with EDs fulfilled the DSM-5 criteria for an ED when evaluated in person by a trained psychiatrist using a structured clinical interview for the DSM-5 criteria [31]. The included diagnoses were: anorexia nervosa (AN), bulimia nervosa (BN), binge eating disorder (BED), and other specified feeding and eating disorders (OSFEDs). Patients’ healthy siblings (HSs) were screened for the exclusion criteria of a personal history of any ED or psychiatric condition. All the participants were recruited via direct invitations. The inclusion criteria for all participants were as follows: between 14 and 40 years old (which is the usual age range of patients treated at the ED unit) with no history of psychotic symptoms or severe medical conditions. A portion of the current sample had already been included in a previous study [29].

All the participants were volunteers and none knew the study’s aim. Participation or refusal did not impact the treatment pathway of the participants with EDs. The study design was approved by the local ethical committees and complied with the Declaration of Helsinki. All the participants—or their parents if they were underage—signed informed consent forms.

### 2.2. Assessment

Each participant filled out a form composed of three self-report questionnaires, while demographic data were collected by researchers. The questionnaires included were the eating disorder examination questionnaire (EDE-Q), the PTSD checklist for DSM-5 (PCL-5), and the symptom checklist-58 (SCL-58).

The EDE-Q is a 28-item self-report questionnaire that evaluates ED symptomatology and psychopathology [32]. It is scored using a 7-point, forced-choice rating scale (0–6), with scores of 4 or higher indicating the clinical range. The questionnaire includes 4 subscales: restraint, eating concern, shape concern, and weight concern. In the present study, the internal consistency was excellent (α = 0.903).

The Italian version of the PCL-5 for the COVID-19 pandemic was used [33]. It is a 19-item self-report measure used to screen and measure posttraumatic symptomatology and reflect the DSM-5 diagnostic criteria of PTSD. Items are scored on a Likert scale ranging from 0 to 4, where higher scores indicate more pronounced PTSD symptoms. As suggested by the literature, a seven-factor structure was applied: intrusion, avoidance, negative affect, anhedonia, dysphoric arousal, anxious arousal, and externalizing behavior. In the present study, the internal consistency was excellent (α = 0.964).

The SCL-58 is a widely known 58-item self-report questionnaire used to evaluate psychiatric symptoms and psychological distress [34]. Items are scored on a 5-point Likert scale where higher scores indicate more pronounced symptoms. It provides five subscales: somatization, obsessive-compulsive, interpersonal sensitivity, depression, and anxiety. In the present study, the internal consistency was excellent (α = 0.980).

### 2.3. Data Analysis

The variables are presented as means and standard deviations or frequencies and percentages, as appropriate. The differences between the patients and HSs were evaluated with various t-tests for independent samples.

Network analysis was performed and included the psychological domain scores for each group, thereby creating a graphical representation of the interconnections between EDs, PTSD, and general psychopathology [35]. The domains are depicted as nodes, while their intercorrelations are represented as lines, or “edges”—the thicker and more saturated the edge, the stronger the correlation. Whenever there is no edge between two nodes, the partial correlation coefficient is zero, meaning that the two variables are independent after controlling for all other variables in the network. To limit the number of spurious connections, we applied the “least absolute shrinkage and selection operator” (LASSO) regularization that shrinks small partial correlations, setting them to zero. The extended Bayesian information criterion (EBIC), a parameter that sets the degree of regularization/penalty applied to sparse correlations, was set to 0.5 in this analysis to avoid most of the spurious edges. To quantify the importance of each node in the network, we then calculated the centrality with three indices: betweenness, closeness, and strength. The betweenness centrality denotes the number of times a specific node acted as a bridge along the shortest path between two nodes, while the closeness centrality measures the number of direct and indirect links between each node and the others. The strength centrality of these inter-node connections is expressed as the degree.

The data management and descriptive analyses were performed using SPSS version 25, and the network analysis was performed using JASP version 0.14.1 statistical software (Department of Psychological Methods University of Amsterdam, Amsterdam, The Netherlands, https://jasp-stats.org/, accessed on 30 September 2022). In order to assess the accuracy of the obtained psychological network, we used the various R packages in version 3.4.4 (R core Team, Vienna, Austria) [36]. To calculate the correlation stability coefficient, we used the bootnet package, which is the maximum proportion of the population that can be dropped so that the correlation between the recalculated indices of the obtained networks and those of the original network is at least 0.7 (it should be above 0.25) [35]. We used the R-package Network Comparison Test (NCT) to test the invariant network structure, the invariant edge strength, and the invariant global strength between the subgroups [37].

## 3. Results

### 3.1. Participants’ Characteristics

A total sample of 240 individuals (222 females and 18 males) was included in the study, with 140 from the Vicenza Unit and 100 from the Brescia Unit. The clinical sample comprised 141 individuals with EDs, while the HS group comprised 99 participants. Of the patients included, 83 were diagnosed with AN, 39 with BN, 8 with BED, and 11 with OSFED. Table 1 shows the participants’ demographic details.

### 3.2. Network Analysis

Figure 1 shows the network structure composed of the EDE-Q sub-scores, the PCL-5 sub-scores, and the SCL-58 sub-scores for both groups. The blue edges indicate positive correlations while the orange edges indicate negative ones.

The plot of the centrality indices of the variables included in the network is represented in Figure 2. The five most interconnected nodes for the ED network were: restraint (M = −1.899), avoidance (M = −1.619), external behaviors (M = −1.136), and dysphoric arousal (M = −1.074). In contrast, the most interconnected nodes in the HS group were: restraint (M = −1.902), dysphoric arousal (M = −1.587), depression (M = 1.388), shape concerns (M = 1.272), and avoidance (M = −1.055). Figure 2 and Figure 3 show the centrality and clustering plots.

The centrality stability coefficient strength of the network of the ED patients was 0.34, while that of the HS group was 0.30, and both are considered acceptable. The groups displayed similar values for the maximum difference in all the edge weights of the networks (M = 1, *p* = 0.856), and the difference in the global strength between the networks was not significant (S = 1, *p* = 0.814). This shows that the internal differences between nodes were not statistically relevant when considering the total networks. However, a visual approach to the networks allowed us to point out the presence of a strong connection between avoidance and anhedonia in the HS group that was not present in the ED patients, with a general greater interconnection between nodes in the HS group compared to the ED patients.

## 4. Discussion

This study aimed to evaluate the relationships between PTSD and psychopathological constructs in patients with EDs compared to their HSs. Our results showed that the most interconnected nodes for the ED network were restraint, avoidance, and external behaviors, while those of the HS network were restraint, dysphoric arousal, depression, shape concerns, and avoidance.

In line with the literature, significant differences emerged between patients and HSs regarding posttraumatic symptoms, with more severe impacts seen in the patients [15]. This is in line with our previous findings [29] and with the pandemic literature that has found a significant increase in posttraumatic symptomatology in patients with EDs [38]. However, an interesting aspect of this study is the absence of network differences between patients and HSs. Inconsistent data was available regarding the vulnerability aspects that characterized the siblings of individuals with EDs [7]. Our results showed that patients and their siblings presented a similar psychological network regarding pandemic effects. This similarity in the relationships between constructs corroborates the idea that there are shared psychological elements between individuals with EDs and their unaffected siblings, with life events that might have had roles in the different health trajectories due to personal coping strategies, emotional regulation abilities, and psychological traits. From this perspective, individual traumatic events might have peculiar roles [7,8,39]. Moreover, the similar effects of shared experiences help to corroborate the idea that extrinsic non-shared elements have a role in the development of EDs, as suggested by psychosocial models and neurobiological studies that elicited the role of personal events in the symptoms severity [13,40]. Indeed, our network analysis showed that despite various levels of general, posttraumatic, and eating psychopathology, there are invariant relationships between psychological constructs in ED patients and their HSs.

The COVID-19 pandemic played a specific and relevant role in the increase and severity of EDs [41,42,43]. However, the reasons for these exacerbations are still unknown, even if authors have proposed the effects of traumatic events—such as isolation, physical illness, changes in routines, and lack of socialization—as explanations [38,44]. Moreover, our data confirmed that eating control (i.e., restraint) represents the core of the connections between psychopathology constructs, with a dysfunctional coping role present during the pandemic in ED patients, as suggested by previous authors [38,45]. Restraint represents a core element in eating psychopathology and has already been related to higher stress levels, negative emotions, and low belongingness, all of which are negative emotional states that patients with EDs seem to have managed with dysfunctional eating behaviors during the COVID-19 pandemic [46,47]. Finally, a relevant role of restraint was also pointed out in the HS network, corroborating the evidence of a central role of food rumination during the pandemic that previous literature reported in the general population [48,49]. However, all these aspects must be evaluated further in future studies.

The absence of a clear difference between networks, however visually present, might also be linked to the psychological variables included. Indeed, different studies have pointed out the transdiagnostic posttraumatic effects in the general population, with emotional regulation being reported as a relevant element [50]. We found a visually connected network in siblings, especially when looking, for example, at the connection between anhedonia and avoidance. This aspect might have been linked to avoidance and thought suppression during the pandemic [51], which might not have been the main psychopathology elements for the ED patients overwhelmed by ED worries [52]. This study has certain weaknesses. Firstly, it was a cross-sectional study measuring posttraumatic symptoms during a single examination session; we cannot, therefore, draw conclusions about the causality of the relationships found. Secondly, we included only a small number of male participants, reducing the generalizability of the results. Thirdly, only self-report questionnaires were used, which renders a potential recall bias or exaggeration in the responses possible.

## 5. Conclusions

Network comparisons provided valuable insights into the relationships between posttraumatic symptoms and psychopathology, showing the absence of a structure difference between ED patients and their siblings in the correlations between psychological variables related to posttraumatic symptoms. Our data highlighted the need for more studies to examine the differences between ED patients and their healthy siblings; this could help corroborate the idea that the interaction between the environment and individual’s characteristics might play a role in patients’ negative psychological responses to experiences and stimuli. Future studies could help determine what elements should be targeted to improve symptomatology in ED patients, especially when such patients are in stressful situations.

## Figures and Tables

**Figure 1 jcm-11-07187-f001:**
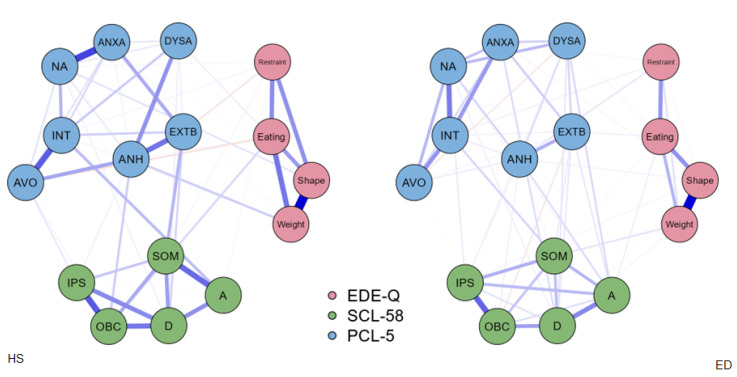
Estimated network structures. The circles represent variables and the lines represent edges (i.e., the association between two variables), and the thicknesses and darkness of lines indicate the weights of the edges. The blue edges indicate positive associations, and the red edges indicate negative associations. Extb: externalizing behavior, anxa: anxious arousal, dysa: dysphoric arousal, anh: anhedonia, na: negative affect, avo: avoidance, int: intrusion, a: anxiety, d: depression, ips: interpersonal sensitivity, obc: obsessive-compulsive, som: somatization, EDE-Q: eating disorder examination questionnaire, SCL-58: symptom checklist-58, PCL-5: PTSD checklist for DSM-5.

**Figure 2 jcm-11-07187-f002:**
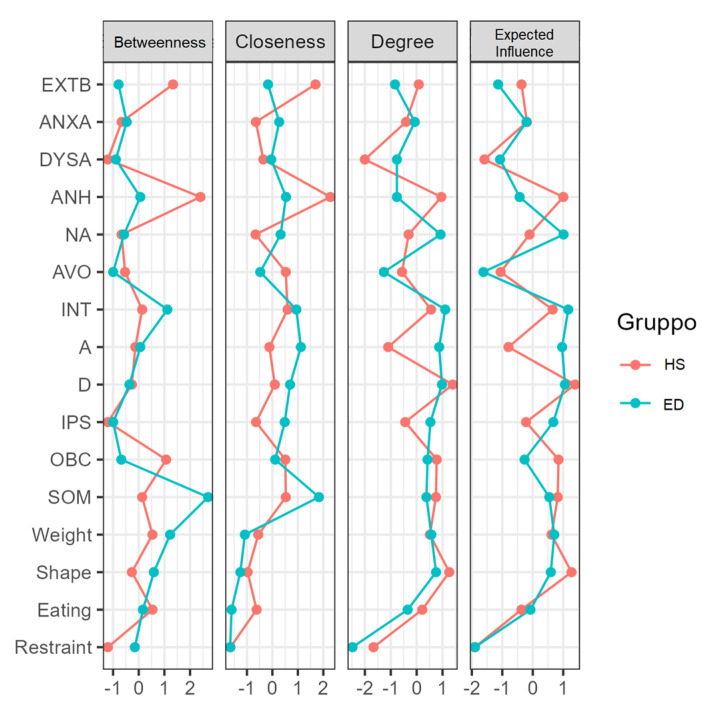
Plot of the centrality indices of the networks depicting the degree, closeness, betweenness, and expected influence of each node. Extb: externalizing behavior, anxa: anxious arousal, dysa: dysphoric arousal, anh: anhedonia, na: negative affect, avo: avoidance, int: intrusion, a: anxiety, d: depression, ips: interpersonal sensitivity, obc: obsessive-compulsive, som: somatization.

**Figure 3 jcm-11-07187-f003:**
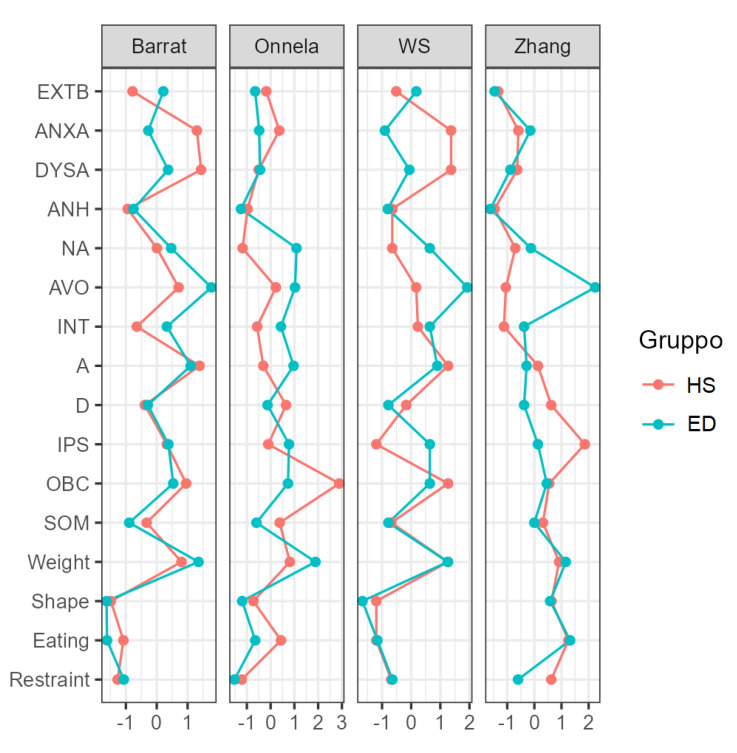
Plot of the clustering indices of the networks. Extb: externalizing behavior, anxa: anxious arousal, dysa: dysphoric arousal, anh: anhedonia, na: negative affect, avo: avoidance, int: intrusion, a: anxiety, d: depression, ips: interpersonal sensitivity, obc: obsessive-compulsive, som: somatization.

**Table 1 jcm-11-07187-t001:** Demographic characteristics of the participants.

	ED n = 141	HS n = 99	*t*	*p*
Age	21.81 (7.19)	23.75 (8.40)	−1.917	0.056
BMI	19.70 (5.62)	21.48 (3.34)	−3.067	0.002
Female (%)	133 (94.3%)	89 (89.9%)	1.643 *	0.302
Years of education	11.58 (3.35)	12.52 (3.37)	−1.790	0.075
EDE-Q				
Restraint	2.83 (1.79)	0.71 (1.25)	10.768	<0.001
Eating concern	3.05 (1.64)	0.74 (1.21)	12.515	<0.001
Shape concern	4.26 (1.66)	1.69 (1.71)	11.649	<0.001
Weight concern	3.55 (1.86)	1.30 (1.42)	10.567	<0.001
SCL-58				
Somatization	1.60 (0.84)	0.79 (0.76)	7.951	<0.001
Obsessive-compulsive	1.53 (0.98)	0.80 (0.80)	6.312	<0.001
Interpersonal sensitivity	1.42 (0.91)	0.74 (0.73)	6.462	<0.001
Depression	1.66 (0.88)	0.86 (0.70)	7.851	<0.001
Anxiety	1.66 (0.94)	0.98 (0.76)	6.195	<0.001
PCL-5				
Intrusion	1.69 (1.02)	1.06 (0.99)	4.739	<0.001
Avoidance	1.79 (1.06)	1.09 (1.00)	5.175	<0.001
Negative affect	1.71 (1.18)	1.01 (0.95)	5.079	<0.001
Anhedonia	2.24 (1.15)	1.33 (1.15)	6.006	<0.001
Dysphoric arousal	1.48 (1.12)	0.91 (0.89)	4.405	<0.001
Anxious arousal	1.68 (1.11)	1.12 (0.97)	4.084	<0.001
Externalizing behavior	1.91 (1.18)	1.37 (1.17)	3.511	0.001

* χ^2^ analysis. ED: eating disorder, HS; healthy sibling, EDE-Q: eating disorder examination questionnaire, SCL-58: symptom checklist-58, PCL-5: PTSD checklist for DSM-5, BMI: body mass index.

## Data Availability

The datasets used and analyzed during the current study are available from the corresponding author on reasonable request.

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
