# Peer review of "A Multicenter Network Analysis Examining the Psychological Effects of the COVID-19 Pandemic on Patients with Eating Disorders versus Their Healthy Siblings"

_jcm, 2022, doi:10.3390/jcm11237187_

Round 1

Reviewer 1 Report

1. Please include the names of the scales in the figure legend for figure 1 (i.e. SCL, PCL, EDE)

2. While there are not overall significant differences between groups, the centrality plots show a couple of items that look significant (anhedonia and avoidance?) It would be worthwhile noting this in the text.

Author Response

  1. Please include the names of the scales in the figure legend for figure 1 (i.e. SCL, PCL, EDE)

R: we thank the Reviewer#1 for the time dedicated to our manuscript. We have now included the names.

  1. While there are not overall significant differences between groups, the centrality plots show a couple of items that look significant (anhedonia and avoidance?) It would be worthwhile noting this in the text.

R: we agreed with the Reviewer. We focused only on one aspect in the first version of the manuscript, but these constructs deserve a discussion. We included this aspect in the manuscript, both in the results and in the discussion.

Reviewer 2 Report

Overall: This paper is very well-written and contributes to the literature on COVID-19 and EDs, as well as EDs and siblings, and the impact of trauma on EDs. With some minor edits

Title:

-        “Examining the Posttraumatic Effects…” While the PCL-5 was used, given that no other post-traumatic symptom questionnaires or interviews were conducted, the title portrays more of a focus on trauma than I read. It may be more accurate to just delete “Posttraumatic” from the title

Introduction:

-        Overall the literature review is concise and appropriate. If the authors wish to continue to focus on posttraumatic stress, I would add a paragraph on the vast literature of trauma and EDs as compared to general population

Line 50 – Reference doesn’t reference siblings; reference may be better placed elsewhere in the sentence.

Line 58 – Reference discussed theories and hypotheticals but a reference that speaks to evidence may be more appropriate, such as the ones you referenced 15-17 (or,   J Devoe D, Han A, Anderson A, Katzman DK, Patten SB, Soumbasis A, Flanagan J, Paslakis G, Vyver E, Marcoux G, Dimitropoulos G. The impact of the COVID-19 pandemic on eating disorders: A systematic review. Int J Eat Disord. 2022 Apr 5:10.1002/eat.23704. doi: 10.1002/eat.23704. Epub ahead of print. PMID: 35384016; PMCID: PMC9087369.)

Lines 65-66: the line needs to be rewritten to add “including” in between pandemic and social isolation so it reads “in the presence of a traumatic event such as the pandemic, including social isolation…”

Lines 75-76: “given different information than the standard comparisons of means” – this is vague and unclear. I would divide the sentence from 74-77 into 2-3 sentences to better introduce the aim of the study with the network analysis approach

Line 76 – I would choose a different word than “diffusion”

Materials and Method:

Participants: Were participants included with any ED, including Unspecified ED? ARFID? BED?

Results:

Participants’ characteristics: Similar to above, it would be helpful to know what ED diagnoses were included in this study, whether the different diagnoses differed in the assessments, and whether this was accounted for in the network analysis

Lines 81-82: Is this clinic an outpatient clinic?

Line 83: “when evaluated in person” – by clinical interview? How were weights obtained – self-report?

Line 86: Were participants recruited for this study only? Did participants self-refer as having an ED? It is mentioned they are volunteers, so assuming that they received no compensation, including treatment?

Discussion:

-        It would be helpful for the reader early in the Discussion to briefly review the most interconnected nodes for the ED and HS.

Line 212 – Noting what the literature has shown in “patients’ sisters” is jarring to read and was not my takeaway from the Maon study cited, nor did this study look only at sisters. I would shift this to read “patients’ siblings” if the authors feel this is still accurate with this sentence.

Lines 210-215 – I had to reread the sentence beginning “The similarities found in this study…” several times and I am still unclear as to its meaning.

Lines 221-222: may be more concise to state “…role in the increase and severity of EDs”. Regardless, ED should have an “s” at the end.

Line 224: I would suggest “isolation” instead of “confinements” (if this was indeed what was meant by the authors) and add “lack of” to “socialization”.

Line 226: Unsure what is meant by “connections between psychopathologies features”

Line 233: Change “been also” to “also been”. For the phrase “…worsening of psychological states, eating and self-control…”, is it meant that eating and self-control are the psychological states? This is unclear and could be reworded or punctuated differently.

Lines 231-237: This hypothesis seems irrelevant to the aims and results of this study.

Conclusions:

Line 247: Instead of “groups” it might be more clear to note “between ED patients and their siblings”

Author Response

Overall: This paper is very well-written and contributes to the literature on COVID-19 and EDs, as well as EDs and siblings, and the impact of trauma on EDs. With some minor edits

R: we thank the Reviewer for the kind words and the time spent reviewing the manuscript.

Title:

-        “Examining the Posttraumatic Effects…” While the PCL-5 was used, given that no other post-traumatic symptom questionnaires or interviews were conducted, the title portrays more of a focus on trauma than I read. It may be more accurate to just delete “Posttraumatic” from the title

R: we agreed with the Reviewer and change the title.

Introduction:

-        Overall the literature review is concise and appropriate. If the authors wish to continue to focus on posttraumatic stress, I would add a paragraph on the vast literature of trauma and EDs as compared to general population

R:we included some more sentences in the second paragraph as suggested by the Reviewer.

Line 50 – Reference doesn’t reference siblings; reference may be better placed elsewhere in the sentence.

R: we agreed. We changed the sentence.

Line 58 – Reference discussed theories and hypotheticals but a reference that speaks to evidence may be more appropriate, such as the ones you referenced 15-17 (or,   J Devoe D, Han A, Anderson A, Katzman DK, Patten SB, Soumbasis A, Flanagan J, Paslakis G, Vyver E, Marcoux G, Dimitropoulos G. The impact of the COVID-19 pandemic on eating disorders: A systematic review. Int J Eat Disord. 2022 Apr 5:10.1002/eat.23704. doi: 10.1002/eat.23704. Epub ahead of print. PMID: 35384016; PMCID: PMC9087369.)

R: we agreed. We included the suggested papers in the text.

Lines 65-66: the line needs to be rewritten to add “including” in between pandemic and social isolation so it reads “in the presence of a traumatic event such as the pandemic, including social isolation…”

R: we thank the Reviewer for the suggestion

Lines 75-76: “given different information than the standard comparisons of means” – this is vague and unclear. I would divide the sentence from 74-77 into 2-3 sentences to better introduce the aim of the study with the network analysis approach

R: we agreed that the aim is better stated now.

Line 76 – I would choose a different word than “diffusion”

 R: we rephrased it.

Materials and Method:

Participants: Were participants included with any ED, including Unspecified ED? ARFID? BED?

 R: we included this information in the methods. We also included in the results the number of participants for each diagnosis.

Results:

Participants’ characteristics: Similar to above, it would be helpful to know what ED diagnoses were included in this study, whether the different diagnoses differed in the assessments, and whether this was accounted for in the network analysis

Lines 81-82: Is this clinic an outpatient clinic?

R: we agreed and included more information about the sample and the centers included. Both are outpatient centers with a day hospital facility.  

Line 83: “when evaluated in person” – by clinical interview? How were weights obtained – self-report?

R: diagnosis were made with the structured clinical interview for the DMS5 criteria. We included this information in the text.

Line 86: Were participants recruited for this study only? Did participants self-refer as having an ED? It is mentioned they are volunteers, so assuming that they received no compensation, including treatment?

 R: they were recruited only for this evaluation. We have now included a statement about the fact that participation or its refusal did not have an effect on their treatment. A paper has already been published with a partial sample, ant it was already reported in the manuscript.

Discussion:

-        It would be helpful for the reader early in the Discussion to briefly review the most interconnected nodes for the ED and HS.

R: we agreed.

Line 212 – Noting what the literature has shown in “patients’ sisters” is jarring to read and was not my takeaway from the Maon study cited, nor did this study look only at sisters. I would shift this to read “patients’ siblings” if the authors feel this is still accurate with this sentence.

R: we agreed.

Lines 210-215 – I had to reread the sentence beginning “The similarities found in this study…” several times and I am still unclear as to its meaning.

R: we splitted and clarified the sentence.

Lines 221-222: may be more concise to state “…role in the increase and severity of EDs”. Regardless, ED should have an “s” at the end.

R: we really appreciated this suggestion

Line 224: I would suggest “isolation” instead of “confinements” (if this was indeed what was meant by the authors) and add “lack of” to “socialization”.

R: agreed.

Line 226: Unsure what is meant by “connections between psychopathologies features”

R: we changed the sentence into “Moreover, our data confirmed that eating control (i.e., restraint) represents the core of the connections between psychopathology constructs, with a dysfunctional coping role during the pandemic in ED patients, as suggested by previous authors”

Line 233: Change “been also” to “also been”. For the phrase “…worsening of psychological states, eating and self-control…”, is it meant that eating and self-control are the psychological states? This is unclear and could be reworded or punctuated differently.

R: we changed the sentence following the point below.

Lines 231-237: This hypothesis seems irrelevant to the aims and results of this study.

R: we agreed. This aspect might be confounding for the readers (and it was totally speculative). So we decided to delete it. We included instead the following sentence:

“Finally, a relevant role of restraint was also pointed out in HSs' network, corroborating the evidence of a central role of food rumination during the pandemic that previous literature reported in the general population. “

Conclusions:

Line 247: Instead of “groups” it might be more clear to note “between ED patients and their siblings”

R: thank you